*Brief communication*:

# Western Europe flood in 2021: mapping agriculture flood exposure from SAR

Kang He[1], Qing Yang[2], Xinyi Shen[1], Emmanouil N. Anagnostou[1]

[1]Department of Civil and Environmental Engineering, University of Connecticut, Storrs, CT 06269, USA
[2]College of Civil Engineering and Architecture, Guangxi University, Nanning, Guangxi, 530004, China

*Correspondence to*: Xinyi Shen (xinyi.shen@uconn.edu)

**Abstract.** In this communication, we present the exposure of agriculture lands to the flooding caused by extreme precipitation in western Europe from 12[th] to 15[th] of July 2021. Overlaying the flood inundation maps derived from the near-real-time RAdar-Produced Inundation Diary (RAPID) system on the CORINE land cover map we estimate a $1.92 \times 10^3$ km$^2$ area affected by the flooding, with 64.37% representing agricultural land. Among the inundated agricultural land, 35.88% of the area is pastures while 33.65% is arable land. Most agricultural flood exposure is found in western France along Rhône River, southern Netherlands along the Meuse River and western Germany along Rhine River.

## 1. Introduction

The heavy precipitation between 12 and 15 July 2021 led to catastrophic floods in western European countries, including France, western Germany, Netherlands, Belgium, and Luxembourg. The flooding caused widespread power outages, infrastructure and crops damages in the affected areas. It is estimated that the loss from the flooding is up to €3 billion [Reinsurance News, 2021]. In addition, 46 people were confirmed dead in North Rhine-Westphalia state in Germany and in the neighbouring state of Rhineland-Palatinate 110 fatalities were confirmed. At least 20 people died following catastrophic flooding in Belgium. The Netherlands, Luxembourg and Switzerland are also affected. Thousands of people had been evacuated from their homes [CNN, 2021; FloodList, 2021]. In the same period, intensive floods occurred in China and the United States. Researchers highlighted that this is an effect of climate change and concluded that the frequency and intensity of such events will increase in a rapidly warming climate [World weather attribution, 2021].

Besides life loss, the flooding in western Europe have also taken a heavy toll on the agricultural sector according to European farmers' association COPA-COGECA. The oxygen supply would be greatly reduced when a corn crop is submerged in water, which greatly reduces or even stops critical plant functions such as nutrient and water uptake [Lauer 2008]. The European Union's crop monitoring unit stated that the exceptionally high rainfall and severe floods would reduce the grain quality in the affected countries [Successful farming, 2021] and had "effectively eliminated" any hope of a successful harvest in these areas [Euractiv, 2021]. Examples of crop damages include crops of grain, rapeseed and flax which have been washed away in Wallonia, Belgium and flood-affected fruit trees along the Meuse River [Eurofruit, 2021]. In widespread crop loss scenarios

like this one, damage assessment is an essential part of flood risk management and flood mitigation, which is also the basis of financial appraisals in the insurance sector [Tapia-Silva et al., 2011]. Even though the impact on the agriculture sector is expected to be severe, the magnitude of the damage is yet to be determined [Agence europe, 2021]. Therefore, it is important to have a quick assessment of the agriculture land exposure to flooding, which will inform crop loss estimates, especially for countries where agriculture plays an important role in the national economy, e.g., France and Germany. Near-real-time (NRT) flood mapping capability from satellite observations is vital to facilitate rapid assessment of flood loss and damage [Shen et al., 2019a].

In this brief communication, we use NRT inundation extents from the near-real-time RAdar-Produced Inundation Diary (RAPID) system combined with CORINE land cover data to depict the flood-affected areas in western Europe, and particularly the agriculture land.

## 2. Methodology

We focus this communication on western Europe, which is mostly affected by the July 12-15 heavy precipitation event. The area extends from 1.5° E to 11.6° E, and 42.9° N to 53.1° N, and encompasses the Netherlands, Belgium, Luxembourg, Switzerland and portions of Germany, France, and Italy. This region is dominated by marine climate with abundant moisture supplemented by Atlantic Ocean. The weather is therefore moist and mild in winter, and moist and cool in summer.

We extract half hourly precipitation data of the event from the Integrated Multi-satellitE Retrievals for Global Precipitation Mission (IMERG) Final Precipitation L3 V06 product with 0.1-degree spatial resolution [https://disc.gsfc.nasa.gov/datasets/GPM_3IMERGHH_06/summary, Huffman et al., 2019]. The IMERG half-hourly Final Run product combines the multi-satellite data for the month with GPCC gauge analysis and thus provides the research-level products for precipitation estimation. We used IMERG data to calculate the maximum hourly precipitation rate and precipitation accumulation between 12 and 15 of July for each grid.

We generate inundation extents in NRT using the RAPID system and archive these maps on Amazon Web Services (AWS) [available at https://rapid-nrt-flood-maps.s3.amazonaws.com/index.html#Global_Flood_Event/Europe_Flood_2021/ ]. RAPID is a fully automated system delineating NRT inundation extents from high resolution (10 m) synthetic aperture radar (SAR) imagery [Yang et al., 2021; Shen et al., 2019a; Shen et al., 2019b]. Specifically, the RAPID system first segments water from non-water pixels by optimizing the threshold and probability density function (PDF) of the water class. Then, it runs a morphology-based procedure to reject false water bodies using rule sets defined at the body level instead of the pixel level. The morphological processing includes two sub modules, water source tracing (WST), and improved changed detection (ICD). WST traces water pixels from known water sources (e.g., rivers, lakes) indicated by a land use map. ICD rejects any water body that is disconnected from a known water source and does not have significantly increased area compared to the dry time. For dry reference, we use information from ground observation and satellite precipitation to determine non-flood period, and image cover that period is select as dry reference. The RAPID requires approximately five dry reference images for each

SAR image sensed on the flood day to reduce the error caused by noise-like speckle. In the third and last processing steps,

RAPID uses multi-threshold compensation and machine learning to further reduce the speckles and strong scatter-caused false negatives. Figure 1 (a) presents an example of inundation delineation by RAPID system in Louhans, France. The CORINE land cover map, shown as Figure 1 (b). The corresponding SAR images sensed on July 16[th], 2021 (Figure 1 (c), flooding period) and images sensed on July 10[th], 2021 (Figure 1 (d), dry reference). To rule out false positives caused by glaciers and snow, we threshold the Height Above Nearest Drainage (HAND) data to mask out permafrost areas in Alps. The HAND used

in this study is obtained from the Multi-Error-Removed Improved-Terrain (MERIT) Hydro Dataset [Yamazaki et al., 2019; Nobre et al., 2011]. Pixels over the Alps where HAND values are greater than 20 meters are removed from the inundated pixels. The threshold is determined by exploring the distribution of HAND for glaciers and perpetual snow recorded in CORINE land cover data and is large enough to avoid the removal of any true positives.

The Landsat-based flood maps are introduced as an independent validation source for the RAPID system. To generate the

flood extent from Landsat, we first acquire surface reflectance image sensed on flooding period from Landsat-8 OLI collection 2 level-2 dataset (Sayler and Zanter 2020), which is available from United States Geological Survey (USGS) Earth Explorer. We then extract the flood extent using the automated water extraction index (AWEI, Feyisa et al. 2014). We calibrate the threshold of AWEI using water pixel samples of high water occurrence. The water occurrence and land use information are extracted from Pekel et al. (2016) and Gong et al. (2019) respectively. Specifically, pixel samples for water are taken from the

persistent water body with more than 90% of water occurrence, and non-water pixel samples are equivalent extracted from the land cover type of cropland, forest, grassland, shrubland, Impervious surface, and bare land. The optimal AWEI threshold is selected as the one that yields the highest F-1 score in segmenting water and non-water pixels.

We obtain the latest land cover map over western Europe from Coordination of information on the environment (CORINE) Land Cover (CLC) inventory data [available at https://land.copernicus.eu/pan-european/corine-land-cover/clc2018 ]. CLC

uses a Minimum Mapping Unit (MMU) of 25 hectares (ha) and a minimum width of 100 meter for linear elements The standard CLC nomenclature includes 44 land cover classes, grouped in a three-level hierarchy. Five main categories used in this study are "artificial surfaces", "agricultural areas", "forest and semi-natural areas", "wetlands" and "water bodies". The detailed description of CORINE program and its nomenclature can be found in https://www.eea.europa.eu/publications/COR0-part1 .

## 3. Results

The spatial pattern of the accumulated precipitation from the July 12-15 heavy precipitation event are shown in Figure 2 (a). Heavy precipitation (peak rate > 20 mm/hr) is observed in western Germany, north-eastern France, norther Luxembourg, south-eastern Netherlands, western Switzerland, and western Italy. The most intensive precipitation (peak rate > 50 mm/hr) is found in western Germany, as well as western Switzerland and Italy over the Alps. Heavier than 150 mm accumulated precipitation is found in eastern France (Châtel-de-Joux, Le Fied), north-eastern France (Plainfaing, Villers-la-Chèvre), mid-eastern

Luxembourg (Echternach and Mersch), western Belgium (Liège), southern Netherlands (Limburg), western Germany (North

Rhine-Westphalia, Rhineland-Palatinate), Switzerland and Italy, which represent an equivalent of two-month precipitation accumulation in these areas. Furthermore, accumulated precipitation is shown to exceed 200 mm in some parts of the region (e.g., western Switzerland, north-eastern France, western Germany).

Figure 2 (b) shows the inundation extents over western Europe, while the four regions where extensive flooded areas are found

from the RAPID inundation map, e.g., the floodplains along Meuse in southern Netherlands, Rhine in western Germany, Rhône in north-eastern France and Arles in south coastal France, are presented as well. The RAPID inundation map shows high consistency with the precipitation map. The total inundated area determined from RAPID inundation map is around $1.92 \times 10^3 \, \text{km}^2$. We find extensive inundated areas in the upstream region of Rhône River where more than 120 mm precipitation fell in 72 hours. The south-eastern France, especially the coastal area, exhibits extensive flood inundation as well, though the

accumulated precipitation in these areas is only around 10 mm. The flooded areas in south-eastern France, shown as Figure 3 (a), are mostly arable land. These areas exhibit clearly dampened backscattering compared with the dry date (Figure 3 (c), June 22) to the flood date (Figure 3 (b), July 16) while their backscattering on the flood date falls into the water category. The croplands labeled as inundated in south-eastern France may be caused by irrigation instead of floods, because the irrigation starts from June 15 in France. As stated in the RAPID algorithm (Shen et al., 2019a), RAPID does not tell the cause of an

incremental area of submerge so the labeled inundation could be caused by irrigation. But authors intend to leave such reasoning to local flood managers or stakeholders because they have better local knowledge and therefore do not think such limitation could cause an issue in disaster response. The total inundated area over France is approximately $1.32 \times 10^3 \, \text{km}^2$. In Germany, the main inundated area is found in the west which is caused by the intensive precipitation (120 mm), along the Rhine River (about 162.02 km$^2$). In the southern Netherlands where more than 100 mm precipitation is observed, the

floodplains of the Meuse, Rhine and Ijssel were largely affected, with a total area of 140.07 km$^2$. We compared the RAPID inundation maps and Landsat-based flood inundation maps (FIMs) for central Netherlands, presented as Figure 4 (a) and (b) respectively. The RAPID and Landsat based FIMs shows high consistency on the flooded areas according to the result of quantitively comparison, with precision, recall, F-1 score and Cohen kappa metrics are 0.88, 0.84, 0.86, 0.86, respectively. In Belgium and Luxembourg, the inundated areas are 116.30 km$^2$ and 1.79 km$^2$, mostly along Meuse River and Sauer River,

respectively. In western Italy, an area of around 50.38 km$^2$ along the Po River is affected by flooding. The flash floods in Switzerland also cause a 130.79km$^2$ inundation.

Table 1 shows the land use fraction in the inundated areas. Among them, 24.16% (463.94 km$^2$) of the land is forested/semi-natural areas. For wetlands and artificial surfaces, the fractions are 5.71% (109.71 km$^2$) and 5.75% (110.49 km$^2$), respectively. The majority, nearly 64.37% ($1.24 \times 10^3 \, \text{km}^2$) of the flood inundated area is from agricultural land. Over inundated agricultural

areas, 35.88% (443.64 km$^2$) is pastures, 33.65% (416.07 km$^2$) is arable land (including non-irrigated arable land and rice fields, 339.04 km$^2$ and 77.03 km$^2$, respectively and 23.34% (288.59 km$^2$) is heterogeneous agricultural areas, which is the sum of complex cultivation patterns (248.04 km$^2$) and land principally occupied by agriculture, with significant areas of natural vegetation (40.55 km$^2$). The remaining 7.10% (87.82 km$^2$) is permanent crops consisting of vineyards (67.11 km$^2$), fruit trees and berry plantations (20.20 km$^2$) and olive groves (0.51 km$^2$).

Specifically, in France, 974.08 km$^2$ of agricultural land cover is affected by the flood. Among those inundated agricultural areas in France, 317.96 km$^2$, 336.67 km$^2$, 233.12 km$^2$ and 86.33 km$^2$ are pastures, arable land, heterogeneous agricultural areas, and permanent crops, respectively. Especially, the non-irrigated arable land in France is severely affected, the area is up to 263.42 km$^2$ which is larger than the sum of inundated non-irrigated arable land in other countries. Besides, the rice fields and vineyards in France are also hit by flood. More than 74.04 km$^2$ of rice fields and vineyards, mainly in the coastal areas, are

inundated. In the Netherlands, 98.97 km$^2$ of agricultural land is inundated, mostly are pastures (50.43 km$^2$), followed by heterogeneous agricultural areas (28.28 km$^2$). The inundated area of arable land (mostly is non-irrigated arable land) in Netherlands is 20.11 km$^2$, while only 0.13 km$^2$ of permanent crops (mainly fruit trees and berry plantations) are affected by flood. In Germany, 88.33 km$^2$ of agricultural land is inundated with 59.28 km$^2$ and 25.13 km$^2$ of these areas being pastures and non-irrigated arable land. The inundation over heterogeneous agricultural areas and permanent crops (including vineyards,

fruit trees and berry plantation) in Germany are estimated at 3.13 km$^2$ and 0.82 km$^2$, respectively. The total inundated areas in Belgium and Italy are 116.30 km$^2$ and 50.32 km$^2$, respectively. In Belgium, the inundated areas of heterogeneous agricultural land, pastures, and arable land were 20.21 km$^2$, 12.57 km$^2$ and 13.63 km$^2$, respectively, while nearly no permanent crop is affected by flood. In Italy, most inundation among agricultural areas is arable land (12.24 km$^2$ of non-irrigated arable land and 2.81 km$^2$ of rice field) and to a secondary effect heterogeneous agricultural area (1.42 km$^2$). Only 0.21 km$^2$ of pastures in Italy

are inundated while 0.06 km$^2$ of permanent crops (vineyards) are affected by flood. In Switzerland, the inundated areas of non-irrigated arable land, pastures and heterogeneous agricultural areas are 4.68 km$^2$, 3.35 km$^2$ and 2.08 km$^2$, respectively. 0.47 km$^2$ of permanent crops, mainly fruit trees and berry plantations, is also found to be affected by flood in Switzerland. No permanent crop is inundated in Luxembourg, the total inundated area in Luxembourg is 1.79 km$^2$, with 0.42 km$^2$, 0.33 km$^2$ and 0.28 km$^2$ of them being heterogeneous agricultural areas, non-irrigated arable land and pastures, respectively.

**4. Closing remarks**

    The unprecedent precipitation heavily damaged the western Europe with catastrophic flooding, causing damage to agriculture which is yet minimally quantified. In this communication, we analyze the inundated area of agricultural land by overlaying the inundation extent derived from RAPID system with CORINE land cover data. The results indicate that the total inundated area over western Europe is about $1.92 \times 10^3$ km$^2$, of which $1.32 \times 10^3$ km$^2$ is in France. Around 64.37% of the flooded area

is agricultural land. Because of the wide impact, we expect that the agricultural productivity in western Europe will be significantly reduced. The mid July when the extreme flood happened is the critical growing season for crops like corn in Belgium, France, Luxembourg and Netherlands, and also the harvest season for wheat in Belguim, France and Germany [Foreign Agricultural Service, 2022]. The quality and production of these crops would be severely damaged. Besides the direct damage to livestock and crops, the soil erosion and sedimentation due to the flood cause significant part of agricultural land

be washed away or become less fertile [Mst et al., 2019; Morris and Brewin, 2014]. In addition, extra costs are needed for pastures and cultivable land to reconstruct and recover.

One of the limitations of this study is primarily inherited from the data sources. The RAPID system in Europe is triggered by IMERG precipitation data, which is a satellite-based precipitation product found to systematically underestimate precipitation in complex terrain areas, such as Alps [Navarro et al., 2019]. In addition, RAPID can not capture the inundation well in the

165 limited floodplains along the small rivers due to the spatial resolution issue, such as those in the floodplains along the Geul river, in southern Netherlands. The irrigation on croplands during the flooding period, like the case in south-east France, may cause uncertainty on RAPID inundation results. The local knowledge from the users can inform RAPID to further improve its accuracy.

With the increasing flood observing capability brought by modern satellite constellations (for example, ICEYE [Ignatenko et

al., 2020]), future directions of this study will include combining the NRT RAPID inundation estimates with developed flood models, crop data and other essential data (soil salinity, crop sensitivity, etc.) to predict flood-damaged cropland areas [Lazin et al., 2021] and associated socioeconomic impact [Gould et al., 2020].

**Author contribution:** KH: formal analysis, writing – original draft and editing. QY: software, formal analysis, data curation. XS and EA: conceptualization, project administration, writing – review and editing.

**Competing interests:** The authors declare that they have no conflict of interest.

**Acknowledgements:** This research was supported by National Science Foundation HDR award entitled "Collaborative Research: Near term forecast of Global Plant Distribution Community Structure, and Ecosystem Function". Kang He received the support of China Scholarship Council for four years' Ph.D. study in University of Connecticut (under grant agreement no. 201906320068).

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

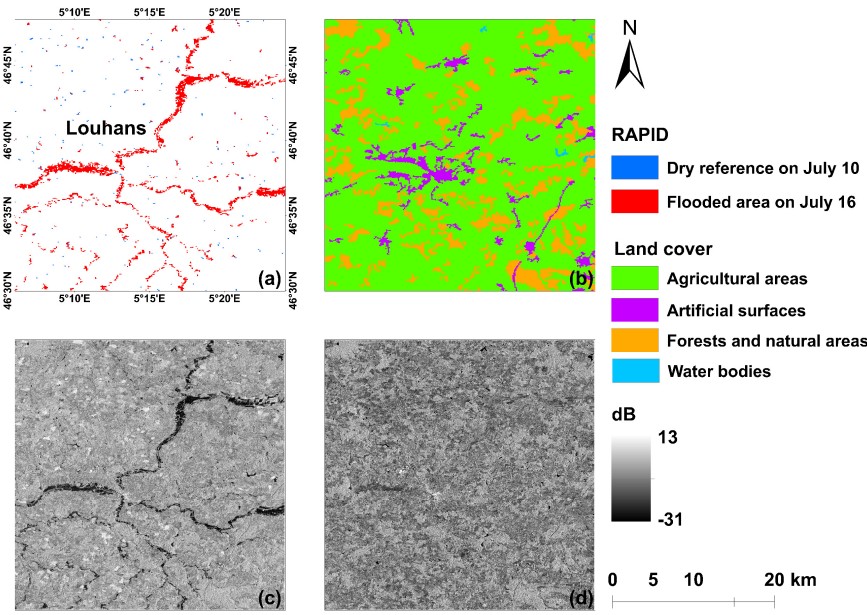

Figure 1. (a) RAPID flood map; (b) CORINE land cover map; (c) and (d) Sentinel-1 SAR image in VH polarization sensed on July 16th and 10th.

245

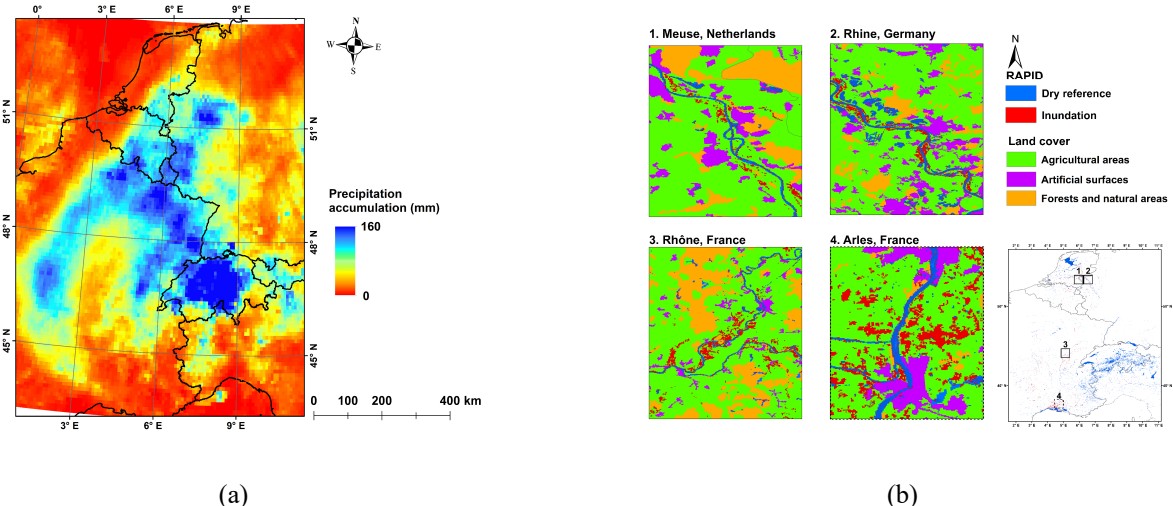

(a)              (b)

Figure 2. (a) Spatial pattern of precipitation accumulation over western Europe from 12th to 15th July, derived from IMERG half-hourly Final Run data. (b) Inundation extents over western Europe from 15th to 18th July, derived from RAPID system.

*Note: The inundation results over south-east France, the regions around the dash rectangle, have uncertainly due to the irrigation during the flooding time.*

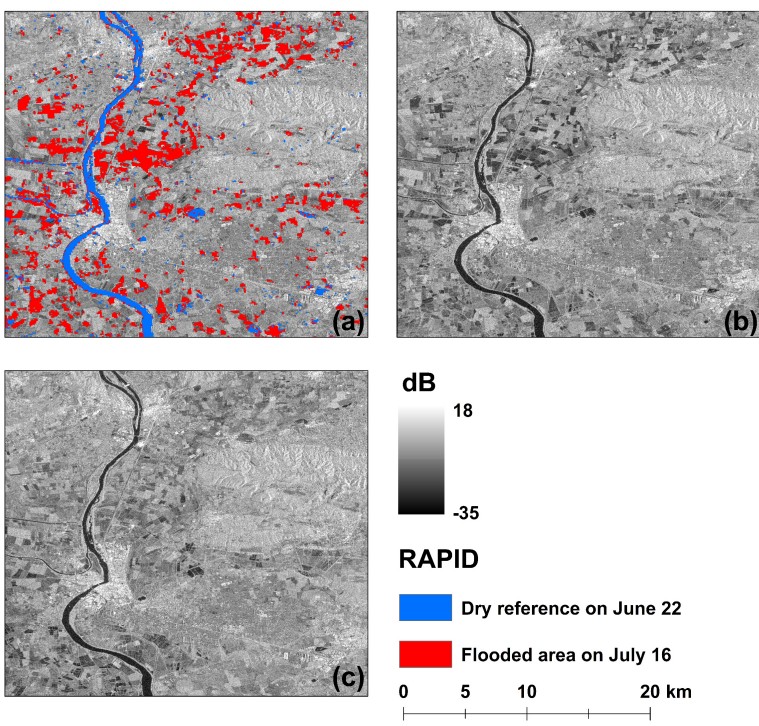

Figure 3. (a) RAPID flood map; (b) Sentinel-1 SAR image in VH polarization sensed on July 16th; (c) Sentinel-1 SAR image in VH polarization sensed on June 22.

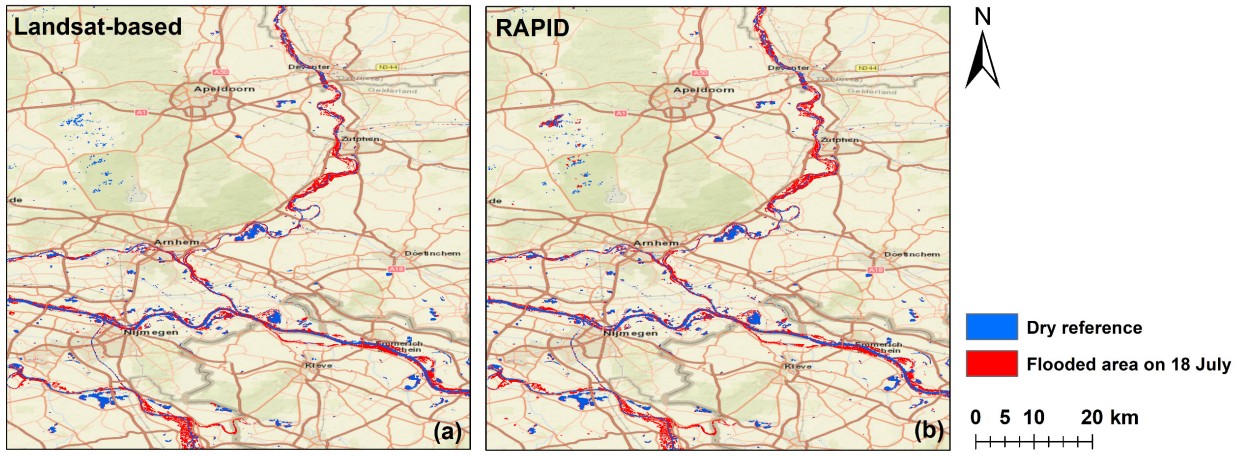

Figure 4. Inundation extent from (a) Landsat-based flood map and (b) RAPID flood map on July 18 in central Netherland.

255

**Table 1. Inundated area of land use grouped by countries over western Europe**

| Inundation area (km²) | France | Germany | Belgium | Netherlands | Switzerland | Luxembourg | Italy |
|---|---|---|---|---|---|---|---|
| Artificial surfaces | 38.19 | 36.49 | 21.78 | 9.42 | 3.88 | 0.08 | 0.65 |
| Agricultural areas | 974.08 | 88.33 | 46.42 | 98.97 | 10.58 | 1.03 | 16.71 |
| Forests and semi-natural areas | 215.84 | 35.47 | 38.46 | 26.98 | 113.56 | 0.68 | 32.95 |
| Wetlands | 90.17 | 1.73 | 9.64 | 5.33 | 2.77 | 0.00 | 0.07 |
| Total | $1.32 \times 10^3$ | 162.02 | 116.30 | 140.70 | 130.79 | 1.79 | 50.38 |

260

