# Peer review of "Western Europe flood in 2021: mapping agriculture flood exposure from SAR"

_Natural Hazards and Earth System Sciences, 2021_

## Author Response (AR1)

**Author's response to reviewers**

The reviewers' comments and questions are marked in red while the author's response to the reviews including a list of all relevant changes made in the manuscript are marked in blue.

**Response to reviewer 1**

*The work presented in this Brief communication includes a concise yet accurate description of the West European flood 2021 and several features on the inundation extent, particularly in agricultural areas. Information that is to date scarcely available. Although it brings limited research novelties, the contribution is timely. Another important point is that it comes with*
*an extensive dataset of satellite derived inundation extent both in georeferences tiff and as images. I'm in favor of publication provided that the following comments are adequately addressed:*

**Respond:** Thanks for your time to review the paper. Your comments are constructive and provide ways to improve the quality of the paper.

Figure 1: Please make country borders thicker. Now they are difficult to see

**Respond:** We have increased the line width of country borders, and changed the color map of precipitation. Now they are clear to see. See figure 2 (b) in the revised paper.

Figure 2: I suggest enlarging the 4 panels on the left and reducing the one on the right. The inundation extent is more informative than the overview map.

**Respond:** Thank you and we agree. Following your suggestion, we have enlarged the inundation extent maps for the four case areas on the left panel while reducing the figure size of the overview map on the right. Now they are in the same size. See figure 2 (b) in the revised paper.

L 83: I suggest avoiding nested parentheses "))"

**Respond:** We have revised accordingly.

For example, from line 115-117, page 4: *Over inundated agricultural areas as Figure 5 (b) shows, 35.9% (443.64 km$^2$) is pastures, 33.6% (416.07 km$^2$) is arable land (including non-irrigated arable land and rice fields, 339.09 km2 and 76.98 km$^2$, respectively) and 23.3% (288.59 km$^2$) is heterogeneous agricultural areas*

L 92: "the Netherlands" I think

**Respond:** We have corrected the typo.

In line 127, page 4: *In the Netherlands, 98.97 km$^2$ of agricultural land is inundated,*

L109-110: This is a sentence for an Abstract or Introduction, rather than for Closing remarks

I think that the information in Figure 3 is already included with more details in Figure 4. I suggest deleting Figure 3.

**Respond:** We have a new start sentence of the closing remarks

From line 145-147 in page 5: *The unprecedent precipitation heavily damaged the western Europe with catastrophic flooding, causing damage to agriculture which is yet minimally quantified.*

The Figure 3 (pie plot) has been removed, and the information of Figure 3 is detailed described in the text.

**Response to reviewer 2**

The brief communication "Western Europe flood in 2021: mapping agriculture flood exposure from SAR" is a very timely contribution for data on the exceptional flooding that took place this year. Particularly the associated data products are a valuable contribution to the knowledge base surrounding this event.

Whilst I support publication of new primary data, particularly in such a timely manner, I do, however, believe there are a couple of things that need to be improved in the communication before full publication is warranted. My main points are on: 1) more detail on the approach used, 2) comparison with other (more local) sources, 3) visualization of the results. Next to these three points, I have some additional minor remarks at the end. Particularly regarding point 2, I am concerned that there may be some artifact in the methodology that heavily impacts the results (see point 2 below).

**Respond:** Thanks for your time and effort to review the paper. Your comments helped us improve the quality of the paper

1. **The methodology is very succinct, but maybe a bit too much so. It is not clear to me as the reader for instance HOW the delineation is done from reading the manuscript. Is it a direct delineation? Or is a comparison made with an image pre-flooding to determine what is flooded and what is normally covered by water. If so, what period is used for reference? This matters as the rivers in question have floodplains that are regularly flooded during winter.**

**Respond**: We have added the details about the delineation method and the images acquired during the flood and used as dry reference in the methodology section. In summary, the RAPID system combines the "direct delineation" using a threshold determination approach, and the comparison using an improved change detection approach.

From line 58 -67 in the revised paper. we added the methodologic details:

*" Specifically, the Radar Produced Inundation Diary (RAPID) system first segments water from non-water pixels by optimizing the threshold and probability density function (PDF) of the water class. Then, it runs an morphology-based procedure to reject false water bodies using rule sets defined at the body level instead of the pixel level. The morphological processing includes two sub modules, water source tracing (WST), and improved changed detection (ICD). WST traces water pixels from known water sources (e.g, rivers, lakes) indicated by a land use map. ICD rejects any water body that is disconnected from a known water source and does not have significantly increased area compared to the dry time. For dry reference, we use information from ground observation and satellite precipitation to determine non-flood period, and image cover that period is select as dry reference. The RAPID requires approximately five dry reference images for each SAR image sensed on the flood day to reduce the error caused by noise-like speckle. In the third and last processing steps, RAPID uses multi-threshold compensation and machine learning to further reduce the speckles and strong scatter-caused false negatives."*

An example of inundation delineation by RAPID, as well as the corresponding SAR images sensed on flood and non-flood date in Louhans, France are presented in Figure 1. The dry reference images are introduced as Figure 1 (d) in the revised paper.

From line 68-70:

*Figure 1 (a) presents an example of inundation delineation by RAPID system in Louhans, France. The CORINE land cover map, shown as Figure 1 (b). The corresponding SAR images sensed on July 16th 2021 (Figure 1 (c), flooding period) and images sensed on July 10th 2021 (Figure 1 (d), dry reference).*

[Figure]

Figure 1. (a) RAPID flood map; (b) CORINE land cover map; (c) and (d) Sentinel-1 SAR image in VH polarization sensed on July 16th and 10th .

**I'm also surprised that around line 50 where the inundation maps are described, there is no reference to the Shen and Yang papers which seem to form the basis for the delineation (judging from the references on the flood maps on AWS). This section really needs more detail for the reader to judge the results adequately.**

We added the citations to the algorithm and system papers from line 52 to 55 in page 2:

*We generate inundation extents in NRT using the RAPID system and archive these maps on Amazon Web Services (AWS) [available at https://rapid-nrt-flood-maps.s3.amazonaws.com/index.html#Global_Flood_Event/Europe_Flood_2021/]. RAPID is a fully automated system delineating NRT inundation extents from high resolution (10 m)*

*synthetic aperture radar (SAR) imagery [Yang et al., 2021; Shen et al., 2019a; Shen et al., 2019b].*

Shen, X., Anagnostou, E. N., Allen, G. H., Brakenridge, G. R., & Kettner, A. J. Near-real-time non-obstructed flood inundation mapping using synthetic aperture radar. Remote Sensing of Environment, 221, 302-315, 2019.

Shen, X., Dacheng W., Kebiao M., Anagnostou, E.N. and Hong Y., 2019b: Inundation Extent Mapping by Synthetic Aperture Radar: A Review, Remote Sensing, 11, 879.

Yang, Q., Shen, X., Anagnostou, E.N., Mo, C., Eggleston, J.R., & Kettner, A.J. (2021). A High-Resolution Flood Inundation Archive (2016–the Present) from Sentinel-1 SAR Imagery over CONUS. Bulletin of American Meteorological Society, 5, E1064-E1079

2.  **The study is done at a relatively large scale to be consistent, which I understand. However, I would really like to see at least some comparison with other estimates. For instance, do the precipitation totals estimated by the authors correspond to some other estimates? These could be from national met offices, or rainfall radars, or**

**other satellite sources, etc. I also noticed that spatially, there is a hotspot of precipitation over the south of Luxembourg. However, the estimates from the Dutch fact finding mission (see below for reference) show this more to the north (northern Luxembourg and eastern Belgium, see Figure 2.2 in the Dutch report). This is based on E-OBS data and in line with the impacts observed in this region.**

**Respond**: We updated the precipitation map using the IMERG half-hourly Final run product which is available now (https://disc.gsfc.nasa.gov/datasets/GPM_3IMERGHH_06/summary). The IMERG Final run product combines the multi-satellite data for the month with GPCC gauge analysis and thus provides the research-level products for precipitation estimation.

From line 48-51 on page 2:

*"We extract half hourly precipitation data of the event from the Integrated Multi-satellitE Retrievals for Global Precipitation Mission (IMERG) Final Precipitation L3 V06 product with 0.1-degree                                                    spatial                                                    resolution [https://disc.gsfc.nasa.gov/datasets/GPM_3IMERGHH_06/summary, Huffman et al., 2019].*

*The IMERG half-hourly Final Run product combines the multi-satellite data for the month with GPCC gauge analysis and thus provides the research-level products for precipitation estimation."*

From the precipitation map in Figure 2, we find precipitation concentrated in western Germany, eastern Belgium, southern Netherlands, Luxembourg, northeastern France and Switzerland. Specially, in the east of Belgium (Liège for example), the precipitation estimate from IMERG
is around 100 mm while more than 200 mm rainfall is observed in these areas from L'Institut royal météorologique (IRM) (https://www.lavenir.net/cnt/dmf20210716_01598142/record-de-precipitations-dans-la-province-avec-plus-de-271-mm-releves-a-jalhay-en-48h    ).    In northeastern France, IMERG estimates total precipitation around 120 mm while according to the observation around 158 mm precipitation fell in the area
(https://meteofrance.com/actualites-et-dossiers/actualites/climat/inondations-catastrophiques-en-allemagne-et-belgique ). In western Germany, the precipitation estimate from IMERG is around 100 mm and according to the report precipitation accumulations over western Germany are averaged 100 to 150 mm.

Overall IMERG precipitation shows high consistency with the observation on the distribution
of precipitation accumulation, though it underestimates the amount in some regions.

[Figure]

**Lastly but crucially, the communication mentions the main inundated area in the Netherlands to be in the north: in the regions of the Markermeer and Ijsselmeer. I know for sure that this is not the case (which is why I want to know more about the**
**methodology) as the Markermeer/Ijsselmeer regions were not impacted at all during the floods. In the Netherlands the impact was way more upstream along the Meuse river (from Belgian border up to Roermond/Nijmegen). My knowledge is mainly on the Dutch situation, but I think it is im perative to check also the other areas for which claims are made.**

**Respond**: In the regions of the Markermeer and Ijsselmeer, we made comparison between RAPID and Landsat-based flood inundation map on 2021-07-18.

From line 103-108 in page 4:

*"In the northern Netherlands where more than 100 mm precipitation is observed, regions near Markermeer and Ijsselmeer, and regions around Hollands Diep and Meuse River are*
*largely affected by the flood, which represents a total area of 140.7 $km^2$. We compared the RAPID inundation maps and Landsat-based flood inundation maps (FIMs) for North Netherlands, presented as Figure 4 (a) and (b) respectively. The RAPID and Landsat based FIMs shows high consistency on the flooded areas according to the result of quantitively comparison, with precision, recall, F-1 score and Cohen kappa metrics are*
*0.8816, 0.8439, 0.8624, 0.8571, respectively."*

For Landsat based flood map, we first acquire surface reflectance image sensed on July 18 from Landsat-8 OLI collection 2 level-2 dataset (Sayler and Zanter 2020), which is available from USGS Earth Explorer. We then extract the flood inundation area by calibrating the pixel threshold based on the automated water extraction index (AWEI, Feyisa et al. 2014).
The pixel samples are selected using the water occurrence (Pekel et al. 2016) and land cover (Gong et al. 2019) data. Specifically, pixel samples for water are taken from the persistent water body with more than 90% of water occurrence, and non-water pixel samples are equivalent extracted from the land cover type of cropland, forest, grassland, shrubland, Impervious surface, and bare land. To determine the optimal threshold, we
delineate water and non-water pixels using all possible thresholds in the valid range of AWEI, and compute F-1 score by comparing to the sample pixels. Finally, we use the threshold with the highest F-1 score to extract flood inundation from Landsat image.

Reference:
Sayler, K., Zanter, K., 2020. Landsat 8 Collection 2 (C2) Level 2 Science Product (L2SP)
Guide. Sioux Falls, South Dakota.

Pekel, J.F., Cottam, A., Gorelick, N. and Belward, A.S., 2016. High-resolution mapping of global surface water and its long-term changes. Nature, 540(7633), pp.418-422.

Gong, P., Liu, H., Zhang, M., Li, C., Wang, J., Huang, H., Clinton, N., Ji, L., Li, W., Bai, Y. and Chen, B., 2019. Stable classification with limited sample: transferring a 30-m resolution
sample set collected in 2015 to mapping 10-m resolution global land cover in 2017. Science Bulletin, 64(6), pp.370-373.

From the comparison, we can find high agreement of inundation extent derived by RAPID and AWEI. The error metrics of precision, Recall, F-1 score, Cohen kappa are 0.8816, 0.8439, 0.8624, 0.8571, respectively.

[Figure]

**I was for instance surprised to learn about the flooding near the coast of Marseille and Montpellier. I did not find any news items on this (though I only looked briefly and don't speak French) and the wiki page also doesn't mention this. So I would urge the authors to check this to make sure it is not the result of an artefact in the method (as I presume the Markermeer/Ijsselmeer probably is), particularly as these are areas that seem to constitute a large portion of the overall results.**

A visual comparison of the SAR images acquired on the flood and dry dates is the most straightforward way to address the reviewer's comment.

In the coast of Marseille and Montpellier, we checked the pre-flood SAR image (June 22, 2021) and the in-flood SAR image (July 16, 2021). The flood areas in south-eastern France are mostly arable land. These areas exhibit clearly dampened backscattering from the dry date (June 22) to the flood date (July 16) while their backscattering on the flood date falls into the water category.

From line 96-100 in page 4:

*"The south-eastern France, especially the coastal area, exhibits extensive flood inundation as well, though the accumulated precipitation in these areas is around 10 mm. The flooded areas in south-eastern France, shown as Figure 3 (a), are mostly arable land.  These areas exhibit clearly dampened backscattering from the dry date (Figure 3 (c), June 22) to the*
*flood date (Figure 3 (b), July 16) while their backscattering on the flood date falls into the water category. These rainfed arable lands might be flooded by overflow from the Rhône river or the raised water table."*

[Figure]

Figure .3 (a) RAPID flood map; (b) Sentinel-1 SAR image in VH polarization sensed on July 16th; (c) Sentinel-1 SAR image in VH polarization sensed on June 22.

**3. The visualization of the results can be improved considerably in the brief communication. Particularly Figure 2 should be improved. Right now no inundation can be seen and even the legend only refers to land-use classes (inundation is not even a class) and it seems more of a land-use map than a flood map. The maps on AWS on the other hand are very informative, so I would put some of those images in the communication.**

**Respond**: We have reworked Figure 2 to improve the visualization in the revised paper. we have enlarged the inundation extent maps for the 4 case areas while reducing the figure size of the overview map.

**I would also pick different areas as the four focus areas. For the Netherlands/Belgium more downstream along the Meuse, for Germany along**

**the Ahr (where most of the impacts where) and in France the communication**
**mentions extensive flooding along the coast (Montpellier/Marseille) and along**
**the Rhone. This would focus the panels on known heavily hit parts/key results.**
**Next to Figure 2, I would propose to include a table with areas affected in the**
**different countries. Now this is listed in text over a couple of paragraphs in the**
**results, but by putting the numbers in a table it would be much easier to**
**compare. Or when the authors feel Figure 4 is sufficient, I would only highlight**
**the main findings from the figure, rather than listing all individual numbers.**

case areas (Meuse river in the Netherland, Rhine river in Germany, Rhone river
and Arles in France) where we find extensive inundation areas area selected and
presented in the left panel of Figure 2 (b). In addition, a summary table concludes
the areas affected in different countries is attached to the right of Figure 2 (b).

[Figure]

| Inundation area (km²) | France | Germany | Belgium | Netherlands | Switzerland | Luxembourg |
|---|---|---|---|---|---|---|
| Artificial surfaces | 38.19 | 36.49 | 21.78 | 9.42 | 3.88 | 0.08 |
| Agricultural areas | 974.08 | 88.33 | 46.42 | 98.97 | 10.58 | 1.03 |
| Forests and semi-natural areas | 215.84 | 35.47 | 38.46 | 26.98 | 113.56 | 0.68 |
| Wetlands | 90.17 | 1.73 | 9.64 | 5.33 | 2.77 | 0 |
| Total | 1318.28 | 162.02 | 116.3 | 140.7 | 130.79 | 1.79 |

Minor remarks:

• The introduction heavily relies on newspaper sources on the event. Whilst these are
of course the first ones to report on it, there have been more specific reports from
the research community as well. In the Netherlands for instance a fact finding
report has been published with an English summary
(https://www.enwinfo.nl/publish/pages/183541/fact-finding-hoogwater-2021-versie-
1-2.pdf)

**Respond**: The detailed reports on the impact of the extreme flood in each country
is cited in the text from Floodlist which is funded by Copernicus. Floodlist aims to
report floods and flooding news since 2008 all over the world.

From line 20-23 in page 1:

*"In addition, 46 people were confirmed dead in North Rhine-Westphalia state in Germany and in the neighboring state of Rhineland-Palatinate 110 fatalities were confirmed. At least 20 people died following catastrophic flooding in Belgium. The Netherlands, Luxembourg and Switzerland are also affected. Thousands of people had been evacuated from their homes [CNN, 2021; FloodList, 2021]."*

CNN: Germany's deadly floods were up to 9 times more likely because of climate change, study estimates, https://www.cnn.com/2021/08/23/europe/germany-floods-belgium-climate-change-intl/index.html , last access: 24 August 2021.

FloodList: Europe, https://floodlist.com/europe, last access: 16 July 2021.

- Some more context can be given in the communication, particularly because it
focusses on agricultural impacts. Most notably: the timing of theses summer floods was crucial as it occurred at the end of the growing season in NW Europe. As a result damage to agriculture can be expected to be relatively high. This is also very rare for NW Europe (Germany, Belgium, Netherlands) where flooding from the Meuse and Rhine rivers is usually during winter [I presume Rhone is similar,
though I am less knowledgeable on that].

**Respond**: We have added the context of growing season period over Europe and discussed the potential impact of flooding on the crops.

From line 150-154 in page 5:

*"The mid July when the extreme flood happened is the critical growing season for crops like corn in Belgium, France, Luxembourg and Netherlands, and also the harvest season for wheat in Belguim, France and Germany [Foreign Agricultural Service]. The oxygen supply is greatly reduced when a corn crop is submerged in water, which greatly reduces or even stops critical plant functions such as nutrient
and water uptake [Lauer 2008]. The quality and production of these crops would be severely damaged."*

Foreign Agricultural Service: Crop Calendars for Europe, https://ipad.fas.usda.gov/rssiws/al/crop_calendar/europe.aspx, last access: 1 March 2022

Lauer, Joe. "Flooding impacts on corn growth and yield." Agronomy Advice. University of Wisconsin. Agronomy Department, Field Crops 28 (2008): 49-56.

- In the communication both precipitation and inundation products are presented. However, the communication seems to focus on the latter one. Some words on how these products are related would be good so the precipitation results don't
feel isolated.

**Respond**: The results of precipitation accumulation and distribution are discussed in the revised paper.

From line 83-89 in page 3:

*"Heavy precipitation (peak rate > 20 mm/hr) is observed in western Germany, north-eastern France, norther Luxembourg, south-western Netherlands, western Switzerland, and western Italy. The most intensive precipitation (peak rate > 50 mm/hr) is found in western Germany, as well as western Switzerland and Italy over the Alps. Heavier than 150 mm accumulated precipitation is found in eastern France (Châtel-de-Joux, Le Fied), north-eastern France (Plainfaing, Villers-la-Chèvre), mid-eastern Luxembourg (Echternach and Mersch), western Belgium (Liège), southern Netherlands (Limburg), western Germany (North Rhine-Westphalia, Rhineland-Palatinate), Switzerland and Italy, which represent an equivalent of two-month precipitation accumulation in these areas. Furthermore, accumulated precipitation is shown to exceed 200 mm in some parts of the region (e.g., western Switzerland, north-eastern France, western Germany)."*

Besides, linkages between precipitation and inundation results are discussed.

From line 93-96 in page 3:

*"Figure 2 (b) shows the inundation extents over western Europe. The RAPID inundation map shows high consistency with the precipitation map. The total inundated area determined from RAPID inundation map is around 1920.26 $km^2$. We find extensive inundated areas in the upstream region of Rhône River where more than 120 mm precipitation fell in 72 hours."*

From line 102-105 in page 4:

*"In Germany, the main inundated area is found in the west which is caused by the intensive precipitation (120 mm), along the Rhine River (about 162.02 $km^2$). In the northern Netherlands where more than 100 mm precipitation is observed, regions near Markermeer and Ijsselmeer, and regions around Hollands Diep and Meuse River are largely affected by the flood, which represents a total area of 140.7 $km^2$."*

• The communication is good to follow, though there are a couple of slightly awkward sentences English-wise (for instance the use of the threshold as a verb in line 51, and permafrost should probably be glaciated in line 52)

**Respond**: We have improved the language throughout the paper in a more academic way.

• Bibliography is not in alphabetical order

**Respond**: We have reordered the bibliography alphabetically.

---

## Author Response (AR2)

**Author's response to reviewers**

The reviewers' comments and questions are marked in red while the author's response to the reviews including a list of all relevant changes made in the manuscript are marked in blue.

**Respond to Review #1**

I'm pleased to see that most of the issues raised in my review were addressed. I've seen considerable changes in comparison to the first version. I've only a few additional comments:

- Figure 2 was modified to address a comment of mine, though I see other issues now. The table in the top-right of the figure is not readable. Also it is sub-optimal to include a table as an image. I'd suggest separating it from the figure and adding it as a table in the main text of in the supplement. In this way the color legend would be moved on top of the map and make it overall more readable.

**Respond:** We have removed the table from Figure 2 and now the table is presented in the Table 1. The color legend of Figure 2 is also removed to the top right of the figure.

Also, the location of the 4 inundated areas should be highlighted in the map on the right.

**Respond:** The locations of the 4 selected inundated areas are marked in the overview map.

[Figure]

**Table 1. Inundated area of land use grouped by countries over western Europe**

| Inundation area (km²) | France | Germany | Belgium | Netherlands | Switzerland | Luxembourg | Italy |
|---|---|---|---|---|---|---|---|
| Artificial surfaces | 38.2 | 36.5 | 21.8 | 9.42 | 3.88 | 0.08 | 0.65 |
| Agricultural areas | 974 | 88.3 | 46.4 | 98.9 | 10.6 | 1.03 | 16.7 |
| Forests and semi-natural areas | 216 | 35.5 | 38.5 | 26.9 | 114 | 0.68 | 32.9 |
| Wetlands | 90.2 | 1.73 | 9.64 | 5.33 | 2.77 | 0.00 | 0.07 |
| Total | $1.32 \times 10^3$ | 162 | 116 | 141 | 131 | 1.79 | 50.4 |

- Use a consistent approach for reporting numbers and their significant digits. 6 is probably too many (e.g., 1236.12 km2). The same applies to "F-1 score and Cohen kappa metrics are 0.8816, 0.8439, 0.8624, 0.8571, respectively.". These would be surely easier to interpret and remember if written as "0.88, 0.84, 0.86, 0.86".

**Respond:** Thanks for your suggestions. The approach for reporting numbers is consistent with 3 significant digits in the revised paper. For example,

**From line 115 to 117 on page 4:**

*"The total inundated area over France is approximately $1.32 \times 10^3$ km$^2$. In Germany, the main inundated area is found in the west which is caused by the intensive precipitation (120 mm), along the Rhine River (about 162 km$^2$)."*

**From line 120 to 122 on page 4:**

*"The RAPID and Landsat based FIMs shows high consistency on the flooded areas according to the result of quantitively comparison, with precision, recall, F-1 score and Cohen kappa metrics are 0.88, 0.84, 0.86, 0.86, respectively.*

- Please make sure that additions fit well within the context and their grammar is correct. In this version it's not always the case. For example, I'm not convinced by the comment on crop oxygenation during floods in the concluding remarks section. This notion was not discovered in this work, it's more suitable for the introduction.

**Respond:** We have moved the comment on crop oxygenation during floods from the closing remarks to the introduction section.

**From line 28 to 29 on page 1:**

**"**The oxygen supply would be greatly reduced when a corn crop is submerged in water, which greatly reduces or even stops critical plant functions such as nutrient and water uptake [Lauer 2008]."

We have checked the grammar and improved the language throughout the paper in a more academic way.

**Respond to Review #2**

I want to thank the authors for making extensive improvements on their manuscript. The method now feels more complete and the figures have much improved. However, I am still hesitant about some of the results and concerned for the presence of artifacts in their results.

Netherlands:

the revised manuscript still reads:

"In the northern Netherlands where more than 100 mm precipitation is observed" -> this is not correct. The intense rainfall was in the south(east) of the Netherlands; not the north. As can also be seen in Figure 2 in the paper.

**Respond:** We apologize for incorrectly describing the precipitation results in the paper. Yes, you are correct, the intense rainfall (> 100 mm) was in the southern Netherlands which we can find from Figure 2 (a).

[Figure]

We have corrected this, from **line 117 to 119 on page 4:**

*"In the southern Netherlands where more than 100 mm precipitation is observed, the floodplains of the Meuse, Rhine and Ijssel were largely affected, with a total area of 141 km$^2$."*

"regions near Markermeer and IIsselmeer, and regions around Hollands Diep and Meuse River are largely affected by the flood" -> as stated in my first review round, the regions near Markermeer and Ijsselmeer were in reality not affected at all. I can't judge where the authors base this finding on but am still afraid there may be artifacts in the methods if this is what they found with the RAPID methodology.

**Respond**: Yes, you are correct. The regions near Markermeer and IJsselmeer are not affected by the flood. we can also know this from the RAPID inundation map that there is no extensive flooded area. We apologize for not correctly describing the inundation results in regions near Markermeer and Ijsselmeer in the last revised paper. We have corrected the description of inundation results over Netherlands in this revised paper.

**Line 117 to 119 on page 4**:

*"In the southern Netherlands where more than 100 mm precipitation is observed, the floodplains of the Meuse, Rhine and Ijssel were largely affected, with a total area of 141 km$^2$."*

[Figure]

Figure 4 shows a part of central Netherlands (not northern) with the rivers Meuse, Rhine, Ijssel, and flooding of floodplains there. This I can confirm happened last summer. But I would phrase it as such (i.e. that the floodplains

of the Meuse, Rhine and Ijssel were affected); as opposed to referencing Hollands Diep, which is a permanent water body).

**Respond:** We have corrected the citation of Figure 4,

*"Figure 4. Inundation extent from (a) Landsat-based flood map and (b) RAPID flood map on July 18 in central Netherland."*

Following your suggestion, we write the inundation results over Netherlands as from **Line 117 to 119 on page 4**:

*"In the southern Netherlands where more than 100 mm precipitation is observed, the floodplains of the Meuse, Rhine and Ijssel were largely affected, with a total area of 141 km$^2$."*

As mentioned in my previous review, the flooding was, by far, the most extensive in the south of the Netherlands (province of Limburg). I would challenge the authors to examine their results around the village of Meersen (confluence of Meuse and Geul) and maybe Valkenburg (along the Geul), which were both severly affected. I can imagine that flooding along the Geul river may not be picked up well as it is a small river with a limited floodplain (much more v-shaped river valley). It could be simply due to resolution issues that this is not picked up; or because the ICD module in the RAPID methodology may discard it if the Geul is not recognized as permanent water body (again due to resolution issues). It is ok if this is the case, but would really like the authors to scrutinize the RAPID methodology on where it does and does not work well. This event gives a great opportunity for that by examining known hit areas. I can imagine that it can pick up very well the flooding of floodplains along major rivers, but less so flooding in smaller upstream rivers with considerably smaller floodplains.

**Respond:** We have checked the inundation results over the province of Limburg, in the south of the Netherlands. We can see the inundated areas in the floodplains along the Meuse River, near Venlo and Roermond, from the RAPID flood map. However, in the regions around the village of Meerssen and Valkenburg, we can not find many inundated areas. We agree with the reviewer that a floodplain which is too small can be ignored because of the resolution issue. We have addressed this as the limitation of RAPID in the closing remark.

The Geul river is not recognized as the permanent water body in the CORINE land cover map. which has a spatial resolution of 100 m. The WST process in RAPID depends on the accuracy of the land use map. **From line 173 to 175 on page 6**:

*"In addition, RAPID cannot capture the inundation well in the limited floodplains along the small rivers due to the spatial resolution issue, such as those in the floodplains along the Geul river, in southern Netherlands."*

[Figure]

South-East France:

I am still very concerned by the results for Southern France. As mentioned by the authors, there was hardly any rainfall there (10mm). I did some searching myself and did not find any mention on floodlist for flooding in that region during that time. What further fuels my concern is the pattern of flooding that is presented in Figure 3. This looks very patchy, unlike the flooding near Louhans and along the Meuse/Rhine/Ijssel rivers (fig1 and fig 4). Given the absence of rainfall, the only flooding I can imagine happening is along the main river that may have received rainfall (far) upstream. But that should not look this patchy and I hardly see flooding linked to the main river. I did notice that also the dry reference (in blue) from June 22 looks very patchy, with many blue spots that in places where on google satellite picture

I see little to no permanent water (for as far as I can judge with just visual comparison). These (in my view) questionable blue spots my result in the RAPID methodology not discarding spots, where in reality it probably should.

Something similar could be the problem with the Ijsselmeer/Markermeer area in the Netherlands which the authors referred to, but I couldn't find the image for that region on the AWS. Btw, the Nijmegen and Roermond figures on AWS there look very good, even the smaller river of the Roer seems to be well captured!

**Respond:** In the southeastern France, the inundation in the floodplains along the main river (Rhône) is due to the heavy precipitation occurred in the upstream of the Rhône River, northeastern France. For the inundation far away from the main river, those patchy areas, they are mainly crop lands (non-irrigated arable land, rice fields, pastures). These arable lands might be flooded by overflow from the river (if they are close to the channel) or the raised water table (close to the coastal), or due to irrigation (since July is the growing season for crops in France, and the official irrigation period is from June 15 up to the French national holiday of August 15). The dry reference SAR images are only available on June 22 when we find these arable lands are totally dry, not been irrigated. For croplands,     RAPID is not responsible to discern      real flood from      irrigated water unless additional information is provided.

**From line 109 to line 114 on page 4**:

*"The south-eastern France, especially the coastal area, exhibits extensive flood inundation as well, though the accumulated precipitation in these areas is only around 10 mm. The flooded areas in south-eastern France, shown as Figure 3 (a), are mostly arable land.  These areas exhibit clearly dampened backscattering compared with     the dry date (Figure 3 (c), June 22) to the flood date (Figure 3 (b), July 16) while their backscattering on the flood date falls into the water category. These arable lands might be flooded by overflow from the upstream Rhône River or the raised water table under the impact of coastal tide.     The croplands labeled as inundated in southern France may also have been irrigated     due to the irrigation starts from June 15 in France."*

https://www.jancisrobinson.com/articles/irrigation-now-official-in-france

[Figure]

RAPID

🟦 Dry reference on June 22

🟥 Flooded area on July 16

For those blue patches in the dry reference, some of them are inland marshes (B and C in the figure) which we can know from the CORINE land cover map. Some crop lands are also identified as the dry reference because of the irrigation water (A in the figure). A few examples are presented below:

[Figure]

**RAPID**

Dry reference on June 22

Flooded area on July 16

Landsat and validation:

The authors seem to have introduced another RS source to estimate flooded areas with, using landsat-based flood inundation maps. However, this is not explained in the methodology (but introduced out of the blue in the results), nor is it clear whether this was an analysis by the authors themselves, or an independent source (there is no citation to anything). Also the reason for doing this (in this one case) is not made explicit. I can guess that it is for valiation, but as this seems to concern a comparison between two estimated flood maps (probably with similar methodology, just different input maps; but

hard to judge as no details are given on the landsat based FIMs), it does not add too much value in my opinion.

**Respond:** Thanks to the reviewer. We did not clearly describe the method used to derive the flood extent from Landsat. Landsat-based flood maps serve as an independent validation source for the RAPID system. So the reference is from a different method and data. Since they highly agree, we can trust the RADID system in places/time where/when we only have Sentinel-1 data. The Landsat-based flood maps is produced using the automated water extract index (AWEI). The process of extraction is summarized below:

**from line 77 to 85 on page 3:**

*"The Landsat-based flood maps are introduced as an independent validation source for the RAPID system. To generate the flood extent from Landsat, we first acquire surface reflectance image sensed on flooding period from Landsat-8 OLI collection 2 level-2 dataset (Sayler and Zanter 2020), which is available from USGS Earth Explorer. We then extract the flood extent using the automated water extraction index (AWEI, Feyisa et al. 2014). We calibrate the threshold of AWEI using water pixel samples of high water occurrence. The water occurrence and land use information are extracted from Pekel et al. (2016) and Gong et al. (2019) respectively. Specifically, pixel samples for water are taken from the persistent water body with more than 90% of water occurrence, and non-water pixel samples are equivalent extracted from the land cover type of cropland, forest, grassland, shrubland, Impervious surface, and bare land. The optimal AWEI threshold is selected as the one that yields the highest F-1 score in segmenting water and non-water pixels."*

References:

Sayler, K., Zanter, K., 2020. Landsat 8 Collection 2 (C2) Level 2 Science Product (L2SP) Guide. Sioux Falls, South Dakota.

Pekel, J.F., Cottam, A., Gorelick, N. and Belward, A.S., 2016. High-resolution mapping of global surface water and its long-term changes. Nature, 540(7633), pp.418-422.

Gong, P., Liu, H., Zhang, M., Li, C., Wang, J., Huang, H., Clinton, N., Ji, L., Li, W., Bai, Y. and Chen, B., 2019. Stable classification with limited sample:

transferring a 30-m resolution sample set collected in 2015 to mapping 10-m resolution global land cover in 2017. Science Bulletin, 64(6), pp.370-373.

Feyisa, G. L., Meilby, H., Fensholt, R., & Proud, S. R. (2014). Automated Water Extraction Index: A new technique for surface water mapping using Landsat imagery. Remote Sensing of Environment, 140, 23-35.

The comparison results show that RAPID flood maps show high spatial consistency with Landsat-based flood maps.

Much more valuable would be validation with independent empirical information such as eye-witness reports (like newspaper reports through floodlist, or social media reports through globalfloodmonitor.org).

**Respond**: Thanks to the reviewer. In Figure 2 (b), we have presented the inundation extent in the some areas where are reported to be severely affected by the flood, like the floodplains along Meuse river in southern Netherlands, floodplains along Rhine river in western Germany and floodplains along Rhône river in northeastern France, in the left panel of the figure.

**From line 103 to line 105 on page 4**:

*"Figure 2 (b) shows the inundation extents over western Europe, while the four regions where extensive flooded areas are found from the RAPID inundation map, e.g., the floodplains along Meuse in southern Netherlands, Rhine in western Germany, Rhone in north-eastern France and Arles in south coastal France, are presented as well."*

we additionally provided the inundation results over areas such as district of Ahrweiler in Rhineland-Palatinate, Euskirchen in in North Rhine-Westphalia, in western Germany; Liège Province and Namur Province in eastern Belgium, where significant damage and many deaths are reported. Overall RAPID can successfully map the inundation area of the flood affected regions.

https://floodlist.com/europe/floods-belgium-july-2021

https://floodlist.com/europe/germany-floods-july-2021

[Figure]

In short: there are several results which I find questionable based on my own knowledge of the event (flooding Ijsselmeer and Markermeer), and based on the results shown by the authors (spotty pattern in SE France). Consequently, I'm afraid there may be incorrect results in certain areas. I would urge the authors to scrutinize their results in order to find the reason

for this. When doing so, the authors can also reflect on when (e.g. when correct dry image is present) or where (e.g. river with substantial floodplain area) the RAPID methodology does very well, and when/where it doesn't perform well. This would be very valuable for the rapid flood mapping community and wider risk management community to learn from.

**Respond:** We have checked the RAPID inundation results over the questionable areas. We have corrected the description of precipitation and inundation results in the Netherlands which are not accurately described in the last revised paper. The extensive inundated areas in south-east France are mainly arable lands. These arable lands might be flooded by overflow from the river if they are close to the river channel, or the raised water table if close to the coast. The spotty pattern in south-east France is result from the inland marshes which can know from the land cover map. Besides, by checking the RAPID inundation results over some severely affected regions from the news and reports, we find RAPID can successfully map the inundated areas over those regions. Overall, RAPID works pretty well in the floodplains along the major rivers, like Rhine, Meuse, Rhone. However, for the limited floodplains along the small rivers, like those along the Geul river in southern Netherlands, RAPID can not capture the flood well due to the resolution issue. We have addressed this issue as the limitation of RAPID system in the closing remarks.

For the inundation results over southern France, we want to emphasize that RAPID cannot tell the difference between flood and the change of ponded status of intermittent water bodies. For instance, if multiple dry dates are used as reference while an irrigated time is used to extract flood, a non-flooded cropland might be identified as inundated. Similarly, an intermittent water body can be identified as a flood area during its high water level period. But we typically do not consider these mistakes as an error of a flood extent extract. RAPID is a near-real time flood mapping system, the inundation results are directly derived from the satellite images. Disaster responders who are using these maps could easily tell these croplands are not impacted by floods with local knowledge. This knowledge can also inform to RAPID to further improve its accuracy. For example, if we know the irrigation plan in advance over some crop fields in southern France, we can mask out these lands.

---

## Author Response (AR3)

**Author's response to reviewers**

The reviewers' comments and questions are marked in red while the author's response to the reviews including a list of all relevant changes made in the manuscript are marked in blue.

**Respond to Review #2**

I want to thank the authors for the adjustments they made to the manuscript with respect to the flooding in the Netherlands. I think this is now much better in line with what is known about the event on the ground.

Respond: We appreciate the reviewer's constructive comments to help improve the quality of the paper.

On the South-East of France I would, however, argue that the authors focus on the wrong explanation. At the moment their results of RAPID are attributed to flooding due to heavy precipitation upstream. Other reasons the authors hypothesize are raised groundwater table or irrigation. I commend the authors on finding the information about irrigation and when it is allowed to do in this region. Contrary to what the authors currently state in the manuscript, however, I strongly believe that irrigation is actually the primary reason for the results of the RAPID analysis. A raised groundwater table due to coastal proximity would be either a more permanent feature, and groundwater in Arlon is too far away from the coast to suddenly rise due to a storm surge (plus I am not aware of a storm at that time). If it was because of river flooding due to upstream rainfall, then the flooded areas should be connected with the main river, as seen in Meuse and Ijssel in the Netherlands. Their results (boxes A, B, C and overview map) clearly display that this is not the case. It is very patchy and the patches are rarely connected to the river. This leads me to strongly believe that RAPID actually picked up on irrigation in the area (which is also impressive and an important lesson). I can't state this with certainty (a dry spell in the region preceeding July 16 would be an indication if that is found). But the explanation the authors currently have – fluvial flooding due to upstream rainfall – is not consistent with their results (patchy, not connected to river); whilst irrigation would be in line with their results.

Respond: We agree with the reviewer that irrigation could be the primary reason leading to the extensive and patchy flooded area in South-East France. Some additional results are provided to back this up.

From line 113 to 115 at page 4:

"*The croplands labeled as inundated in south-eastern France may be caused by irrigation instead of floods, because the irrigation starts from June 15 in France. As stated in the RAPID algorithm (Shen et al., 2019), RAPID does not tell the cause of an incremental area of submerge so the labeled inundation could be caused by irrigation. But authors intend to leave such reasoning to local flood managers or stakeholders because they have better local knowledge and therefore do not think such limitation could cause an issue in disaster response.*"

Shen, X., Anagnostou, E. N., Allen, G. H., Brakenridge, G. R., & Kettner, A. J. Near-real-time non-obstructed flood inundation mapping using synthetic aperture radar. Remote Sensing of Environment, 221, 302-315, 2019.

First, by overlaying the RAPID flood map with the global map of the irrigation areas of coarse resolution, obtained from Food and Agriculture Organization (FAO) of the United Nations (https://www.fao.org/aquastat/en/geospatial-information/global-maps-irrigated-areas/latest-version/ ), we find high spatial correlation between the patchy flooded area and the area equipped for irrigation over south-east France. While irrigation is permitted from June 15 up to August 15 in France, it is possible that these regions in south-east France have been irrigated and therefore are detected by the RAPID system as flood during the flooding period (July 15 to July 21, 2021). In the original paper of RAPID, authors concluded one of the limitations is that RAPID does not differentiate the cause of an incremental area of submerge.

[Figure]

Figure 1. RAPID flood map and global map of the irrigation areas over south-east France.

Second, from one image on May 30, before the official irrigation time, we do not find those patchy water bodies in south-east France, which appears on the dry reference image, used in this study, on June 22 (Figure 2 (a)). Based on the precipitation map from IMERG data, the total precipitation is only around 10 mm over the areas which cannot be the cause of patchy water bodies appearing on June 22. The irrigation started from June 15 could be an explanation. Similarly, the extensive patchy flooded areas in the RAPID flood map on July 16 (Figure 2(b) might also be caused by the irrigation over these regions.

[Figure]

[Figure]

Figure 2. (a) Dry reference on May 30 and June 22 and accumulated precipitation from June 18 to 21 obtained from IMERG; (b) Dry reference on June 22 and RAPID flood map on July 16 and accumulated precipitation from July 12 to 15 obtained from IMERG;

Overall, I think the manuscript has important lessons to learn on how flooding can, and cannot, be detected from remote sensing with a nice application to a very recent and relevant event. However, I do not feel comfortable publishing results that claim widespread flooding in SE France – of a scale that would definitely have reached (inter)national media – without auxiliary arguments/data to back this up. Particularly when it does not follow, in my opinion, logically from their own results (patchy pattern) and there is a more plausible explanation available (irrigation).

Respond: Thanks to the reviewer for the confirmation of this study. As responded to the last comment, based on the additional evidence provided above, we think that irrigation in south-east France since June 15 might be the primary reason leading to the widespread flooding found in the RAPID flood map. We have addressed this in the revised manuscript as quoted in the last response

Besides, we have added caption text on Figure 2 to address the triggering reason of the inundation results over south-east France.

[Figure]

Figure 2. (b) Inundation extent over western Europe from 15th to 18th July, derived from the RAPID system.
*Note: The inundation results over south-east France, the regions in the dash rectangle, might not be caused by the flood due to the irrigation during the flooding time.*

In closing remarks, we also stated this case as one of the limitations in the current RAPID system.

From Line 175 to 177 at page 6:

"*The irrigation on croplands during the flooding period, like the case in south-east France, may cause uncertainty on RAPID inundation results. The local knowledge from the users can inform RAPID to further improve its accuracy.*"

**Respond to editor**

**editor**

Thank you for the submission of your revised paper "Brief communication: Western Europe flood in 2021: mapping agriculture flood exposure from SAR" to NHESS.

In the previous review round and revised manuscript, you have provided clear and convincing rebuttals and improvements to most of the points raised by the reviewers. However, one of the reviewers still has a major concern with regards the results (and the explanation of the results) for flooding in SE France. I agree completely with these concerns. The floods that your results show are of huge size, yet I cannot find any news sources or other information on this, and your results do not provide any other source to back this up. As stated by the reviewer, a flood did occur around Arlon in 2003, but showing a very different pattern. I am therefore unable to accept the manuscript for publication. I would like to give you one opportunity to revise or rebut this review, either providing strong evidence that this very large (and what would have been impactful flood) actually occurred, or providing another strong argument for a possible other mechanism (and discussion of what this means for the reliability of RAPDI), such as that suggested by the reviewer.

I also had one small point remaining. In the previous review, reviewer 1 suggested using a consistent approach for reporting numbers and their significant digits. Whilst you have partially implemented this, there are still some inconsistencies (e.g. in table 1 sometimes no value behind decimal point, sometimes one, and sometimes two). Please also amend these if you decide to submit a revised manuscript.

If you decide that you would like to submit a new version, I look forward to seeing the next version of your manuscript which I will then send out for further review to the previous reviewer (if they agree) or new reviewers.

Please be aware that this is most likely the last possibility for you to change and improve the manuscript.

Respond: We have responded to the comments from Reviewer #2 with additional results provided in the response letter. We agree with reviewer #2 that the irrigation occurred in south-east France during the flooding period could be the primary reason leading to the widespread flooding found on the RAPID flood map. Some evidence is provided to back this up, and the manuscript is revised accordingly.

Besides, we have adjusted the approach to reporting the numbers. Now the numbers in text and table are consistent with two decimal digits.

---

## Author Response (AR4)

**Author's response to reviewers**

The reviewers' comments and questions are marked in red while the author's response to the reviews including a list of all relevant changes made in the manuscript are marked in blue.

5 I want to thank the authors for making the adjustments to the manuscript. I now feel much more confident in the findings they present. My only remaining minor comments would be some small textual things (mainly relating to too many decimals/significant digits) and elaborating slightly on the rationale for irrigation (in results) and on the limitations (resolution/irrigation) they found 10 in this study (where they come from and how to address in future).

Respond: We appreciate the reviewer's time and effort to improve the quality of the paper, we have revised the paper accordingly. The detailed revisions are presented below.

Abstract:

15 1.92 x 10^3 km2 -> 1920 km2

Respond: We have revised the approach for reporting numbers in the abstract and in the rest of the manuscript.

in line 11 at page 1:

"*we estimate a 1920 km$^2$ area affected by the flooding*"

20

Western France along Rhone River (Rhone is not in western part of France! check how you describe the region later in the manuscript)

Respond: Thanks for pointing out this. Rhone River runs through south eastern France. We have revised this in the abstract as well as those in the 25 manuscript.

From line 13 to 14 at page 1:

"*Most agricultural flood exposure is found in eastern France along Rhône River, southern Netherlands along the Meuse River and western Germany along Rhine River.*"

30

line 28-29: I would remove the inserted line about oxygen supply, creates more confusion than it adds.

Respond: We have deleted this sentence in the revised paper.

line 105: Put a full stop after "[...] in north-eastern France." Then make "Arles in south coastal France is presented as well." a seperate line (also to seperate the inundated areas from this other area of interest more clearly).

Respond: We have revised it accordingly.

From line 101 to 103 at page 4:

"*Figure 2 (b) shows the inundation extents over western Europe, while the regions where extensive flooded areas are found from the RAPID inundation map, e.g., the floodplains along Meuse in southern Netherlands, Rhine in western Germany, Rhône in north-eastern France are displayed. The inundation extent over Arles in south coastal France is presented as well.*"

line 113-115: I miss the line of argumentation about the pattern of the flooded areas from RAPID (patchy, not linked to river) as an indication that this is probably an irrigation signal, and not a flooding signal.

Respond: The explanation of the patchy pattern of the flooded areas in south-eastern France can be found in the revised paper.

From line 109 to 110 at page 4:

"*The flooded areas in south-eastern France whose pattern is primarily patchy, not linked to rivers, are mostly arable land, shown as Figure 3 (a).*"

and

From line 113 to 117 at page 4:

"*The croplands labeled as inundated in south-eastern France may be caused by irrigation instead of floods, because the irrigation starts from June 15 in France. As stated in the RAPID algorithm (Shen et al., 2019a) RAPID does not tell the cause of an incremental area of submerge so the labeled inundation could be caused by irrigation. But authors intend to leave such*

*reasoning to local flood managers or stakeholders because they have better local knowledge and therefore do not think such limitation could cause an issue in disaster response."*

line 117: would say about 162 km2 (remove the .02)

Respond: We have revised it accordingly.

line 120: remove the decimals again, so area of 140 km2

Respond: We have revised it accordingly.

Line 124-156: you now quote too many decimals / significant digits (sometimes 5 significant digits and 3 decimals!). The results are not that precise in terms of uncertainties, plus it is a lot less easy to grasp for the reader. I would remove the decimals everywhere. Only when number is below 10 you can use 1 decimal (so 2 significant digits then).

Respond: Following your suggestion, we have modified the approach for reporting numbers throughout the paper. In the revised paper, the decimals are removed when numbers are greater than 10, and 1 decimal is used when number is below 10.

Line 164: Also not a fan of x10^3 notation. Just say 1920 km2 and 1320 km2 as that is easier for reader to interpret.

Respond: We have revised it accordingly.

line 172: You start with 'one of the limitations', but actually list three limitations (IMERG precipitation, smaller tributaries, irrigation). I would make that more explicit. I would also spend maybe a few more words on this; indicating where the limitation comes from (i.e. spatial resolution for Geul) and how it can be addressed (higher resolution land-use; expert assessment of flood patterns and local knowledge).

Respond: We have rewritten the limitations of the study in the revised paper.

From line 174 to 180 at page 6:

*"The findings of this study have to be seen in light of some limitations. The first is that RAPID system in Europe is triggered by IMERG precipitation data, which is a satellite-based precipitation product found to systematically underestimate precipitation in complex terrain areas, such as Alps [Navarro*

*et al., 2019]. The more accurate precipitation data over Europe is recommended to provide more reasonable inundation results over these areas. The second limitation concerns spatial resolution. RAPID can not*

95 *capture the inundation well in the limited floodplains along the small rivers due to the spatial resolution issue, such as those in the floodplains along the Geul river, in southern Netherlands. Higher resolution data, including satellite imagery data and land use data, could help RAPID system to map the flooded area along the small rivers. Lastly, the irrigation on croplands during*

100 *to the flooding period, like the case in south-east France, may cause uncertainty on RAPID inundation results. The local knowledge from the users can inform RAPID to further improve its accuracy."*